# Sustainable and Security Focused Multimodal Models for Distance Learning

**Vacius Jusas** [1] , **Rita Butkiene** [1] , **Algimantas Venčkauskas** [1] , **Šarūnas Grigaliūnas** [1] , **Daina Gudoniene** [1,*] , **Renata Burbaite** [1] and **Boriss Misnevs** [2]

1   Faculty of Informatics, Kaunas University of Technology, 44249 Kaunas, Lithuania; vacius.jusas@ktu.lt (V.J.); rita.butkiene@ktu.lt (R.B.); algimantas.venckauskas@ktu.lt (A.V.); sarunas.grigaliunas@ktu.lt (Š.G.); renata.burbaite@ktu.lt (R.B.)
2   Department of Software Engineering, Transport and Telecommunication Institute, 1019 Riga, Latvia; misnevs.b@tsi.lv
*   Correspondence: daina.gudoniene@ktu.lt

**Abstract:** The COVID-19 pandemic has forced much education to move into a distance learning (DL) model. The problem addressed in the paper is related to the increased necessity for the capacity of data, secure infrastructure, Wi-Fi possibilities, and equipment, learning resources which are needed when students connect to systems managed by institutional, national, and international organizations. Meanwhile, there have been cases when learners were not able to use technology in a secure manner, since they were requested to connect to external learning objects or systems. The research aims to develop a sustainable strategy based on a security concept model that consists of three main components: (1) security assurance; (2) users, including administration, teachers, and learners; and (3) DL organizational processes. The security concept model can be implemented at different levels of security. We modelled all the possible levels of security. To implement the security concept model, we introduce a framework that consists of the following activities: plan, implement, review, and improve. These activities were performed in a never-ending loop. We provided the technical measures required to implement the appropriate security level of DL infrastructure. The technical measures were provided at the level of a system administrator. We enriched the framework by joining technical measures into appropriate activities within the framework. The models were validated by 10 experts from different higher education institutions. The feasibility of the data collection instrument was determined by a Cronbach's alpha coefficient that was above 0.9.

**Keywords:** infrastructure; distance learning; security models; education

## 1. Introduction

The disruption of learning processes disrupted by the COVID-19 pandemic has involved a radical transformation of education and training, and one of the sectors undergoing dramatic digital transformation globally is higher education [1]. The sudden forced closure of face-to-face teaching activities has led many academics and many students into "unfamiliar terrain" due to the need to adapt swiftly to total distance learning (DL) settings [2]. This sudden change has required universities to evolve toward DL in record time, implementing and adapting the technological resources available and involving professors and researchers who lack innate technological capacities for DL.

DL requires large resources of computers, information, and communication channels. Additionally, DL faces other challenges that are as follows: (1) organization of DL processes associated with practical and laboratory work, (2) skill testing and evaluation using information and communication technology, (3) forecasting of system load, (4) cybersecurity, and (5) data protection issues. The infrastructure of DL, which consists of three sections: management and governance, physical infrastructure, and logical infrastructure [3], is indispensable in the learning processes.

The emergence of disruptive innovation is a time of risk and uncertainty, but it is also a time of opportunities, bringing talent and innovation to the education system [4]. The questionnaire at King Saud University, Saudi Arabia [5] revealed that the success factors of DL named by the students and faculty members differ. The importance of technical skills, effective time management, individual differences, and support is fostered by the technology infrastructure of the DL environment [6]. When each of these pillars is equally prioritized in fully DL delivery, ultimately the best-equipped students succeed in their course from orientation through to graduation.

The use of technology helped educators to overcome the issue of DL during disaster times, but the educators argued that robust IT infrastructure is a prerequisite for DL [7]. The infrastructure needs to be strong enough that it can provide unrestricted services during a pandemic [8]. Huang et al. [9] underlined that reliability and sufficient availability of, for example, communication infrastructure, learning tools, and digital learning resources are of utmost importance in such severe situations.

Since DL is being improved and supported by technical innovations and infrastructure, many users of innovative infrastructure are not experts in using it, therefore, it is necessary to have a support staff and an established structural system for successful entrance to the DL market [10].

The problem addressed in the paper is related to the increased necessity for the capacity of the data, secure infrastructure, Wi-Fi possibilities, and equipment, which are needed learning resources when students connect to systems managed by institutional, national, and international organizations. Meanwhile, there have been cases when learners were not able to use technology in a secure manner since they were requested to connect to external learning objects or systems.

The research question is "How to assure secure and effective DL by developing security concept models consisting of three main components: (1) security assurance; (2) users, including administration, teachers, and learners; and (3) DL organizational processes".

One of our co-authors is employed as a security manager of a whole university network. All of his suggestions came from a practical point of view. So, he practically knows the viability of the proposed models and their technical merits to ensure the security of DL infrastructure.

This paper is structured as follows: The related work is reviewed in Section 2. Section 3 presents a problem formulation and a model of DL infrastructure. A security concept model and implementation framework are presented in Section 4. Results and discussion are provided in Section 5. The conclusions of the paper and the limitations of the research work are discussed in Section 6.

## 2. Literature Review

Literature reviews include 42 articles from Scopus, Web of Science, and databases related to the topic. Screening was performed in two phases. The first phase was used for screening titles and abstracts and the second phase is screening full texts. We used a reference management system for literature resources collecting for citing the most appropriate papers related to the topic.

Tyagi and Verma [11] focused on security in sustainable education and distance learning. The authors [11] presented sustainable education as an educational approach aimed at entrenching in students, schools, and communities the values and motivations to act for sustainability now and in the future—in one's own life, in their communities, and on a worldwide platform.

However, the authors of the paper present the Sustainable Multimodal Model for DL as the model suggesting to educational organizations systematic and consistent ways to effectively implement teaching and learning in the worldwide platforms.

Gaiveo [12] presents the security of the computer-based information systems, that links with the preservation of the information that is supported by those systems, controlling information and systems collection, treatment, use, support, and accesses.

The authors of the paper describe the Security Focused Multimodal Models for Distance Learning as the models for educational organizations to guide them on: (1) security assurance; (2) safe and effective management of users, including administration, teachers, and learners; and (3) DL organizational processes.

We provide a review of research works considering issues of DL infrastructure and security. We firstly review the research works devoted to the development of DL infrastructure. We then review the research works that analyze security threats to the DL infrastructure and suggest ways mitigate the security threats.

Ergüzen et al. [13] suggests improving the technological infrastructure of DL through a trustworthy hardware platform-independent remote education laboratory. The reason for this suggestion is that the students are not able to acquire the costly software needed for studies. A platform-independent remote laboratory suitable for computers, smartphones, and tablets has been developed for DL students in information technology. To access the virtual laboratory a security layer containing student-specific information and security measures was used. The developed laboratory had additional benefits as follows: (1) no need for installation on students' own computers; and (2) provides a way for students with weak computers to use server power for all transactions. The students using the newly developed laboratory got a 12.89% higher average mean score than the students using traditional methods in a web programming course.

Moore and Fodrey [14] present a model of a DL technology infrastructure. The model consists of four components: systems, objectives, evaluation, and personnel. Each of these components is required for any DL technology infrastructure. It is an IT division-level framework. There is not a specific order for this model on how projects may be initiated. For the component of the system, there are two aspects of interest within the system's component. The first is what delivery method is used to create instructional content, and the second is what tool will be used to deliver this content. It is critical to establish clear learning objectives and to align them with the technology tools that are selected and implemented. Learning objectives must be firstly considered, then only tools selected and evaluated. There are two parts of evaluation within this model. The initial evaluation of the selected tool and the continuous evaluation must occur after the implementation. Finally, the division will need to determine if it has personnel who can support this new tool for the faculty members and for live-event support. The main shortcoming of this model is that the model is provided at the IT division level. The authors of the model recognize that a technology infrastructure plan is needed for e-learning leaders. However, planning is not an activity of the model.

Thomas [15] clusters the infrastructure of DL into four layers from bottom to top: internet, hardware, software, and rules and regulations. Three groups of actors interact with the infrastructure of DL. These groups are as follows: institutions, individual instructors, and individual learners. Associating the layers of DL to actors enables a better understanding of the specific challenges. These challenges vary for different actors. For example, broadband internet is usual for the institution, but it can be a problem for the individual instructors and learners. The same is valid for hardware and software. However, Thomas [15] did not consider the security issues related to the infrastructure of DL.

García-Peñalvo [16] define a reference framework for introducing eLearning practices into face-to-face education. The proposed framework consists of seven layers. The basic level of the framework is an infrastructure that is divided into three sections: management and governance, physical infrastructure, and logical infrastructure. The government of technologies is an essential factor for the success of DL. The physical infrastructure to support DL must cover all the different needs concerning physical equipment. The logical infrastructure includes the software components and users with experience who are also part of DL infrastructure. Security is the fifth layer in this framework. The security is considered together with ethics and privacy issues. Privacy of individuals must be respected and based on General Data Protection Regulation (GDPR) [17]. However, García-Peñalvo [16] provided no details on how to solve the security issues.

Not only the infrastructure of DL, but also the security of DL platforms is a key success factor of DL [18,19]. DL platforms are important for billions of users and they rely on them to perform routine activities. The user-friendliness of the graphical user interface and their constant availability made them vulnerable. Therefore, it is important to secure web applications from attacks. Bhatia and Maitra [20] analyzed all the open-source e-learning platforms available in the market today to test their vulnerability against attacks. All the analyzed e-learning platforms had severe vulnerabilities. Moodle, which is the most popular e-learning platform [21], was not an exception. Bhatia and Maitra [20] propose a model for ensuring the security of e-learning platforms. The proposed security model is posed on a two-fold holistic view: a hierarchical approach and a distributed approach. The hierarchical approach consists of the following layers from top to bottom: system administrator, instructors, and learners. The distributed approach keeps separate security models of each element in the e-learning platform. The advantage of the hierarchical approach is centralization. The advantage of the distributed approach is scalability. The implementation of the security model could be done in four steps: (1) define security policy; (2) implement security policy within an e-learning system; (3) launch constant monitoring; and (4) react to an ongoing attack. The last step is invoked by the third step in case of need. A regular update is needed to the security model since threats change. The main disadvantage of the proposed security model is that the model was presented only as a collection of ideas. The security model of the e-learning system was not validated by experts.

Husain and Budiyantara [22] analyzed the effect of the control security and privacy on the attitude and behavioral intentions of e-learning users. The results of the investigation indicate that the control of security and privacy has a significant influence on the attitude and behavioral intention of e-learning users.

During the pandemic, the number of online resources drastically increased. However, the number of cyber-attacks increased, as well [23]. Khan et al. [24] identified ten deadly cybersecurity threats during the COVID-19 pandemic. They are as follows: (1) DDoS attacks; (2) malicious domains; (3) malicious websites; (4) malware; (5) ransomware; (6) spam emails; (7) malicious social media messaging; (8) business email compromise; (9) mobile threats; and (10) browsing apps. These threats are oriented to the general e-community rather than e-learners. It is important to know the most common cybersecurity threats since e-learners are part of the general e-community. Khan et al. [24] noticed also that the most widely used online conferencing tool, Zoom, faced massive criticism because the default settings of privacy and security are not adequately secure.

Ali and Zafar [25] stated that an institution providing e-learning must implement robust measures to protect sensitive participants' data against loss or unauthorized use. For this purpose, Ali and Zafar [25] presented a conceptual model of the information security and privacy factors related to e-learning. The factors are as follows: (1) data evaluation; (2) policies; (3) legislation/regulation; (4) architecture; (5) integration; (6) training; and (7) risk analysis. The authors analyzed all the factors and made propositions for every factor. The most compelling proposition is for the first factor. This proposition says that the participation of all stakeholders in the data evaluation step will enhance the security and privacy of e-learning technology. No reason was provided why this specific order of enumeration of the factors was chosen. The ordering of factors raises some doubts. Ali and Zafar [25] also provided recommendations for the implementation of the proposed conceptual model.

Ran et al. [26] asserted that identity verification of the user of the DL platform should not terminate at the login process, but it must proceed as the user is connected to the platform. Therefore, Ran et al. [26] developed an identity authentication model based on the private cloud. The model is based on multi-fold security approach and it provides an authentication methods repository. The repository includes classical methods and behavioral validation methods that are as follows: email verification, two-step verification (login + SMS), Captcha test, face recognition, fingerprint identification, speech recognition,

and keystroke recognition. After the initial login, the process of constant verification starts. The main component of this process is face recognition. However, the method faces constraints of network bandwidth and computer hardware.

Nita and Mihailescu [27] proposed a secure framework for e-learning platforms using attribute-based encryption applied in cloud computing. The framework consists of the following components: methods and criteria, education and training, e-learning users, improved access mechanism, security layer, cloud, authorities, and owners. Improved access mechanism means attribute-based encryption. However, the proposed framework is not well-structured since it connects the different types of components. Moreover, neither validation nor experiments are provided using the proposed framework.

Amo et al. [28] analyzed how learning management systems (LMSs) store and process personal data. The authors established that these data are stored unencrypted, and these data are easily accessible to many users of LMSs. Therefore, the LMSs are vulnerable to the loss of sensitive information and such information storing is not in line with GDPR. To comply with GDPR, LMSs apply a simple solution: everything or nothing. If you do not agree with terms and conditions, you do not have access. The authors suggested solving the problem to use an access matrix. The suggestion to store the personal data in encrypted form was also provided. The suggested solution was implemented for the LMS Moodle.

Amo et al. [29] proceeded with the earlier investigation on GDPR implementation in the LMS and studied the possibility for the students to be anonymous in the LMS since GDPR delegates such a right. The LMS does not have such a function. The questionnaire was carried out among learners and educators. The educators did not contradict to teach the anonymous students. Therefore, the authors implemented an add-in for the LMS Moodle. The add-in is called "Protected Users", which allows hiding of the learner's identity. The add-in is freely available on GitHub.

Caviglione and Coccoli [30] noticed that online learning is an interplay among social, educational, and technological aspects. They suggested a model to identify and classify security threats and vulnerabilities of e-learning frameworks in smart cities. The model is called a holistic one, but no proof is provided. The model is divided into three spaces: infrastructure, data, and learner. A training and technological awareness of individuals is a prime countermeasure for learner space. No countermeasures are proposed to fight security threats in the infrastructure and data spaces.

Mahmood [31] presented an agent-based framework for providing the security and privacy of the cloud-based E-learning. An architecture of the cloud-based E-learning usually consists of five main layers: hardware resource layer, software resource layer, resource management layer, service layer, and business application layer. The service layer is comprised of the three services: SaaS (software as a service), PaaS (platform as a service), and IaaS (infrastructure as a service). The framework is introduced just to SaaS service.

Alexei and Alexei [32] observed that use of cloud computing (CC), learning management systems (LMS), and video conferencing applications (VCA) has become the mainstream for conducting distance learning. They presented a review of security threats to these three types of applications and provided common recommendations to secure CC, LMS, and VCA. These common recommendations include classifying information, implementing access policies at the application or resource level, updating systems, and using cryptographic protocols.

We provide a summary of the main features of the discussed related works in Table 1.

**Table 1.** Summary of related works.

| Research Work | DL Infrastructure | Security Issues |
|---|---|---|
| Ergüzen et al. [13] | Hardware platform-independent remote laboratory. | VPN layer ensures security. |
| Moore and Fodrey [14] | Model consists of four components: systems, objectives, evaluation, and personnel. | Not considered. |
| Thomas [15] | Four layers of the DL infrastructure from bottom to top: internet, hardware, software, and rules and regulations. | Security measures are managed by IT departments. No details are provided. |
| García-Peñalvo [16] | Three sections of DL infrastructure: management and governance, physical infrastructure, and logical infrastructure. | Security is fifth layer in the framework. No details are provided. The security is considered together with ethics and privacy issues. |
| Bhatia and Maitra [20] | Not considered. | Hierarchical and distributed approaches. The hierarchical approach consists of the layers from top to bottom: system administrator, instructors, and learners. The distributed approach keeps separate security models of each element in the e-learning platform. |
| Ali and Zafar [25] | Not considered. | Conceptual model of the information security and privacy consists of factors in the specific ordering: (1) data evaluation, (2) policies, (3) legislation/regulation, (4) architecture, (5) integration, (6) training, and (7) risk analysis. |
| Ran et al. [26] | Not considered. | Identity authentication model using multi-fold security approach based on classical and behavioral validation methods. |
| Nita and Mihailescu [27] | Not considered. | Secure framework using attribute-based encryption consists of the components: methods and criteria, education and training, e-learning users, improved access mechanism, security layer, cloud, authorities, and owners. |
| Amo et al. [28,29] | Not considered. | GDPR compliant personal data storing in LMS Moodle. |
| Caviglione and Coccoli [30] | The model is divided into three spaces: infrastructure, data, and learner | A training and technological awareness of individuals is a prime countermeasure for learner space. |
| Mahmood [31] | Not considered. | The framework is introduced just to SaaS service. |
| Alexei and Alexei [32] | Not considered. | Common recommendations are provided to secure cloud computing, learning management systems, and video conferencing applications. |

We can conclude, observing Table 1, that none of the authors of the reviewed papers have demonstrated a systematic view of the problem of the security of the DL infrastructure. Moreover, the reviewed authors demonstrated different understanding of DL infrastructure. In the next section, we will present our view on the DL infrastructure.

## 3. Problem Formulation and the Model of DL Infrastructure

Based on the literature review, we developed the requirements for the models of the DL infrastructure and security:

1. The model of DL infrastructure must include all the components required for distance learning.
2. The security concept model should cover: (1) IT infrastructure security profile; (2) levels of IT security; and (3) secure and reliable DL infrastructure framework.

3. The plan for the implementation of the DL infrastructure framework, which joins levels of IT security and IT infrastructure security profile, should be provided in the form of a matrix.
4. The security concept model must include the protection of personal data.
5. The security concept model must include the management of copyright and licenses for digital content and software in education.

The IT infrastructure of the organization is a set of hardware, software, technical, communication, information, organizational, and technological tools that ensure proper functioning and management. The IT infrastructure includes a combination of various applications, databases, servers, disk arrays, and network equipment, and provides users with access to information sources. IT infrastructure is a technological component of any service that ensures the delivery of the considered service following agreed rules and procedures. During the pandemic, the range of participants of DL has significantly increased, their IT skills vary greatly, and the available tools for remote work (RW) differ in terms of technical parameters: workplace equipment has different capacity, and the capacity of communication channels is often insufficient.

Therefore, carefully selected, developed, and configured IT infrastructure is essential to ensure reliable and secure DL and RW (Figure 1). Next, we present the components of the model of DL infrastructure.

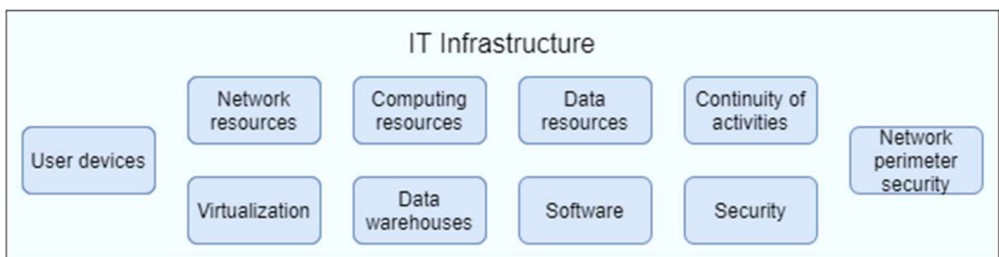

**Figure 1.** The model of DL infrastructure.

User devices constitute an exceptionally large spectrum of computing devices used by the participants of the DL process. This includes smartphones, tablets, and personal and desktop computers. The capabilities of these devices differ in ensuring secure and reliable DL and RW. This imposes some restrictions on the components of DL infrastructure.

Network resources include modems, routers, Wi-Fi devices, and communication lines. These resources together with user devices must ensure secure and reliable user connection with computing and information resources. To accomplish the mentioned goal the devices must use secure communication protocols (HTTPS, SSL/TLS, virtual private network, and others) and they must ensure the required bandwidth.

Virtualization is usually used in modern technologies, especially in cloud computing. Virtualization enables more effective, more flexible, and more secure use of resources. This is useful for the organization of DL and RW.

Computing resources (processors and memory blocks) are one of the most important resources, especially during the pandemic when requirements for the resources significantly increase.

Data warehouses store the digital contents of e-learning, the generated contents (individual assignments, projects, control works, and others) by e-learners, the private information of participants of DL, and others. During the pandemic, this resource is the most important since the requirements for it drastically increase.

Data resources, digital contents of e-learning, the organizational data of DL, and private data of participants of DL are critical for the process of DL. Therefore, high requirements are imposed on these resources.

The software encompasses the operational software, software of e-learning platforms, educational programs, and others. Software is one of the main components of DL infras-

tructure. The software is solely responsible for the ability to present e-learning material to the users of DL, for the ability of the users to communicate online and offline during the process of DL.

Continuity of activities is a set of regulations and rules that are responsible for the operation of an institution during various conditions.

Security is a set of organizational, legal, and technical regulations that are responsible for ensuring the secure operation of DL infrastructure.

Network perimeter security (firewalls, incident detection software) is a set of tools to ensure the secure operation of inner IT infrastructure.

## 4. Security Concept Model and Implementation Framework

### 4.1. Security Concept Model of DL Infrastructure

Three components take part in the DL process [16]. They are as follows: people, IT infrastructure, and organizational processes. A security model of DL infrastructure must ensure the security of the entire DL ecosystem. Figure 2 presents a security concept model that consists of security assurance, people, and organizational processes.

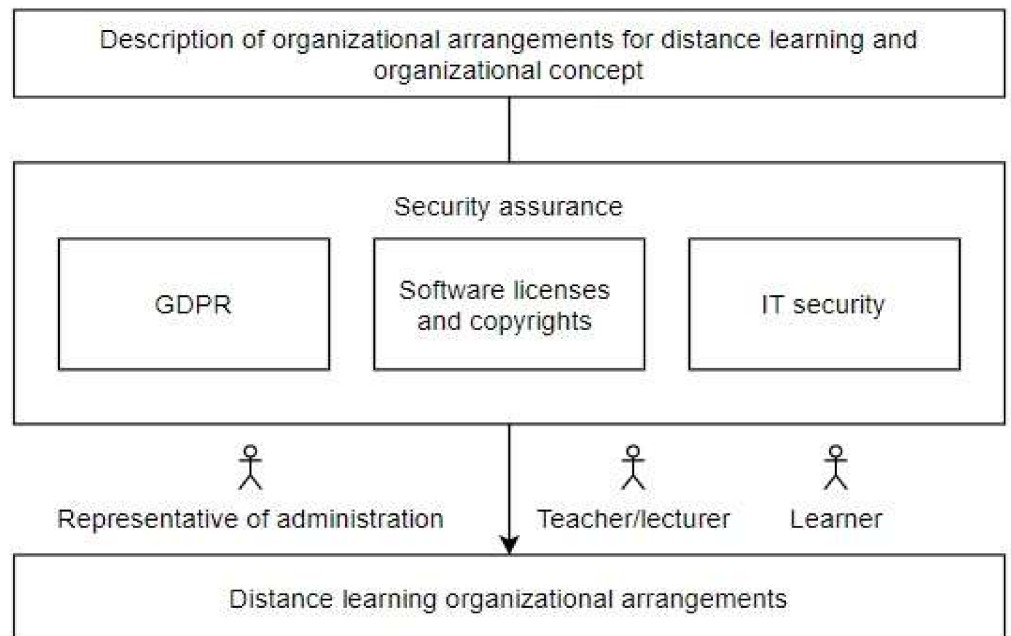

**Figure 2.** Security concept model of DL ecosystem.

Organizational processes of DL were considered in our previous research work [33]. They include methods of e-learning and educational technologies to support these methods.

In the security assurance, we distinguish three the most important components (Figure 2):

- Security of IT infrastructure;
- Protection of personal data (GDPR);
- Copyright and licensing of digital content and software.

Next, we will consider the implementation of these three components.

### 4.2. Security Profile of IT Infrastructure and Levels of IT Security

To implement security of IT infrastructure we present a security profile of IT infrastructure (Figure 3).

For the implementation of security of IT infrastructure, we used the standard ISO 27001 [34], which sets out requirements for an information security management system so that the organization can assess risks and put in place appropriate controls to protect confidentiality, integrity, and availability of information. IT infrastructure is also

associated with the control of access to associated technologies. This is defined by the standard COBIT 5 [35] which is one of the most popular IT management methodologies developed and supported by the organization of ISACA (Information Systems Audit and Control Association).

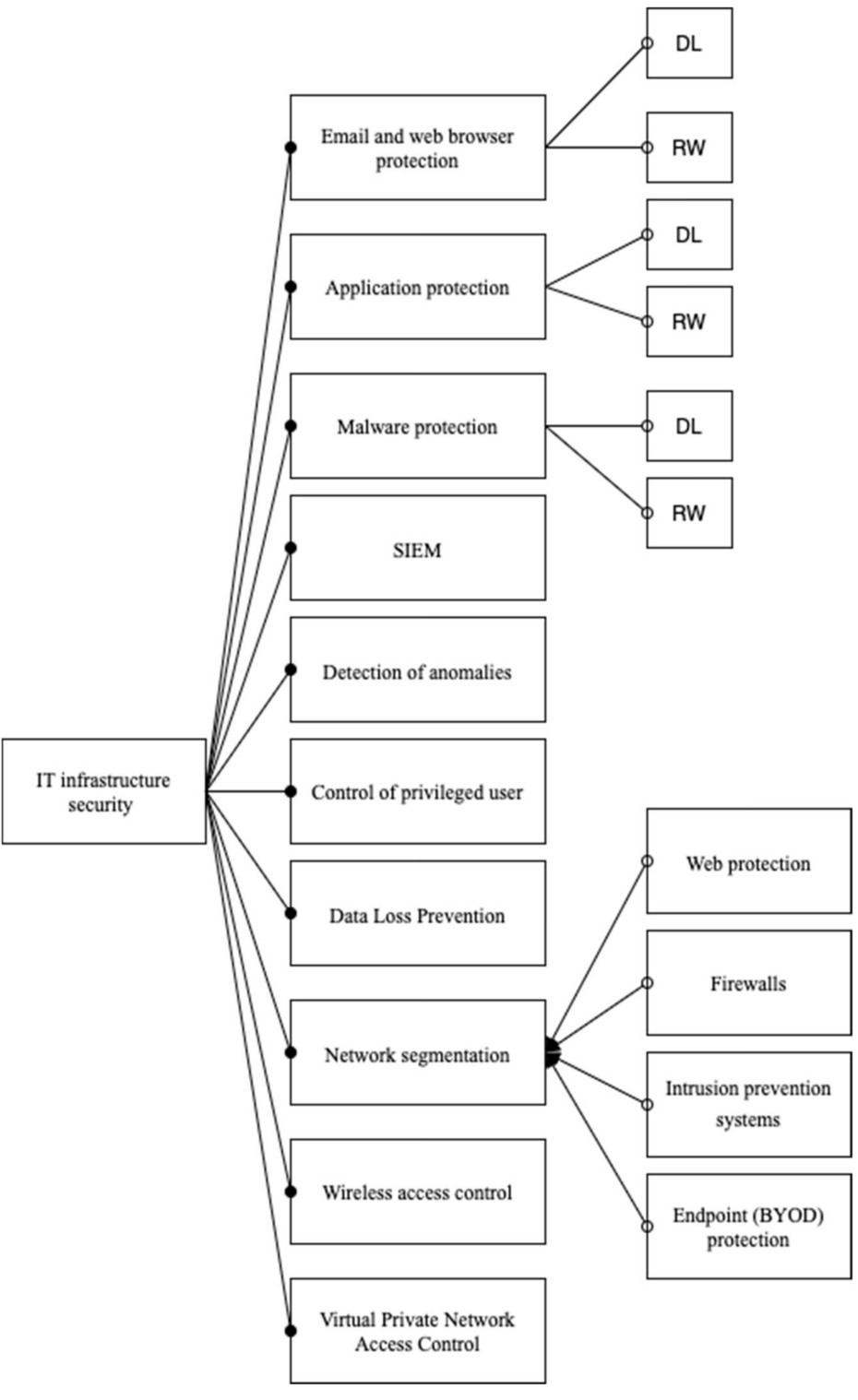

**Figure 3.** The security profile of IT infrastructure.

When dealing with IT infrastructure solutions, it is necessary to assess the risks and information security aspects (Figure 3):

1.  Threats (external and internal): Every employee and endpoint are potential points of entry into the network. It is possible exploitation of vulnerabilities, a combination of spam, fraud, malicious URLs, and social engineering that are easier than ever to detect, automate, and deploy.
2.  Risks associated with sensitive data: organizations handle many types of sensitive data, including protected learner records and data that is to be exported. The protection of sensitive data requires a good understanding of the nature of the information, knowing its location, knowing how it is generated, transmitted, shared, stored, processed, and ultimately deleted. This security profile of IT infrastructure deals with the identification, documentation, implementation, and control to protect high-value assets and sensitive information throughout its lifecycle.
3.  Information security covers three main aspects:
    *   Information confidentiality—protection of information against unauthorized disclosure;
    *   Information integrity—protection of information against unauthorized or accidental alteration of data and/or information;
    *   Access to information—ensuring that information is available when it is needed by designated users of information.

Next, we present all the components of the security profile of IT infrastructure.

Email and web browser protection is oriented to the security flaws related to the user actions. The malicious persons using different fraud strategies persuade the users of email to share sensitive information or to download the malicious programs into the network. The email protection enables the identification of spam. It can be used to identify dangerous e-letters, block attacks, and prevent sharing of sensitive data.

Application protection must enable the protection of all the application programs that can be related to the security of organization networks and security metrics. Application programs are the usual target of hackers.

Security information and event management (SIEM) tools enable a selection of data from different resources to be put into single storage for quick action in the case of need.

Anomaly detection is not a straightforward action. It can be quite difficult to find anomalies in the operations of an organization network since no one knows how the network operates in the anomaly-free mode. A careful investigation is needed to learn anomaly-free network operation mode. The available network anomaly detection tools allow analysis of the network operations. They can establish an early warning when a violation of the network operations is detected.

Data loss prevention (DLP) technologies and policy usually enable the protection of users who do not intend to use secret data improperly, and will not lose sensitive data in the network. The human factor is the weakest part of the chain of network security. Therefore, safeguards are needed to prevent either malicious or unintentional but harmful actions of the users.

Network segmentation allows assigning of the appropriate safeguard to the particular traffic in the network since traffic from different resources requires different protection. Such network partition enables application of the specific protection to each type of traffic.

Web protection is a segment of network security. Web protection is a generic term, which encompasses all the tools and measures that the organization must enable to ensure the secure usage of the internet in the inner network of the organization. Such protection does not allow the use of browsers as a means to invade the organization's network.

A firewall is a segment of network security. The goal of a firewall is to protect boundaries between an organization's network and the internet. A firewall is used to manage the network traffic and to block access to undesirable traffic.

Intrusion prevention systems (sometimes called intrusion detection systems) are a segment of network security. They constantly read and analyze the network traffic to notice as quickly as possible various attacks on the network and to react to them. These systems are founded on the basis of the known attacks [36] to recognize the known threats.

Endpoint (BYOD) protection is a segment of network security. Organizations allow users to use their owned computing devices. This is called bring your own device (BYOD). However, user-owned devices usually do not possess such strong protection as an organization's computers. Therefore, they become an easy target for hackers. Therefore, the endpoint protection adds a defensive layer between remote computing devices and organization networks, for example, a virtual private network (VPN).

Wireless access control is a separate component in the security profile of IT infrastructure since wireless networks are less secure than traditional networks. Therefore, additional measures are needed to secure wireless networks.

Virtual private network access control is used to enable authenticated communication between the endpoint device and secure organization networks. For remote VPN access authentication, either IPsec or Secure Sockets Layer protocol is used to create an encrypted communication channel that other interesting parties would not be able to access the transmitted data.

Security education is an important component of the security profile of IT infrastructure, as well. It provides four major benefits to organizations: (1) improve employee behavior; (2) increase the ability to hold employees accountable for their actions; (3) mitigate the liability of the organization for an employee's behavior; and (4) comply with regulations and contractual obligations.

The security profile of IT infrastructure can be implemented at different levels of security (Figure 4).

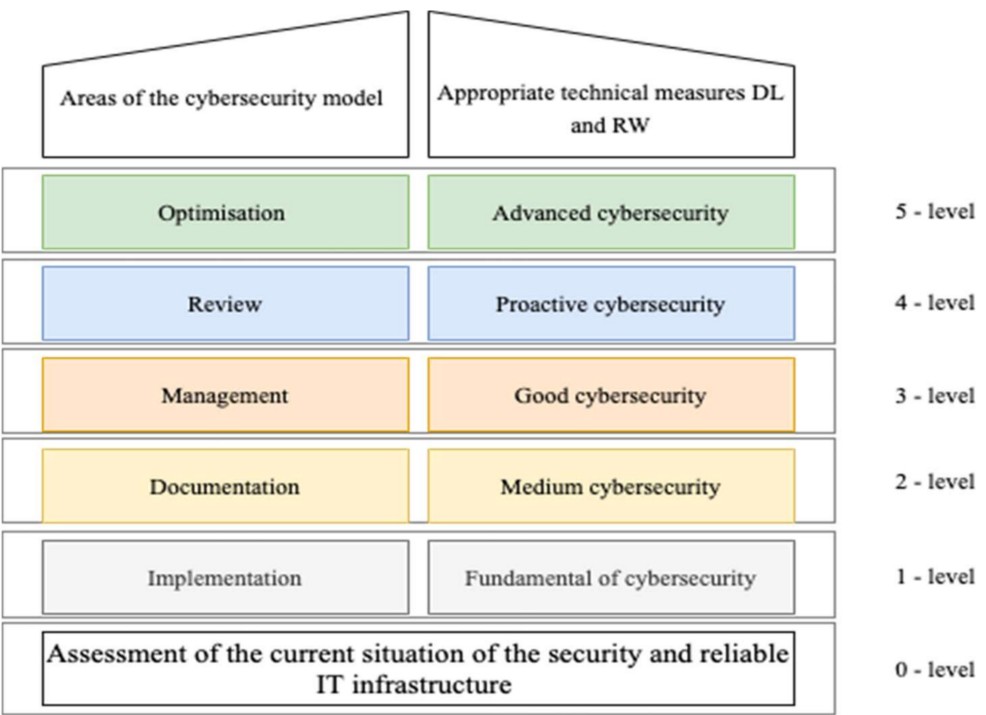

**Figure 4.** IT security levels.

The security levels within an organization can be ensured by appropriate technical measures that are enumerated in the security profile of IT infrastructure. To ensure that the areas of the cybersecurity model and the level of cybersecurity are sufficient, it is necessary to assess the situation of the organization's existing IT infrastructure. It is a ground level of cybersecurity.

### 4.3. The Framework to Ensure the Secure and Reliable DL Infrastructure

The framework to ensure a secure and reliable DL infrastructure is based on a Lean [37] methodology. It is one of the methodologies to implement agile methods. This methodology is a process improvement methodology based on reducing resources and improving the efficiency of providing them.

The first step of the Lean methodology is the most important one that is an assessment of the current security situation of DL infrastructure to eliminate wasting of resources. For this purpose, it is necessary to carry out an analysis and identify specifics of the protection level for those working and learning remotely. When all the information concerning the current situation is collected, we can start a framework that opens a never-ending loop (Figure 5). The start of the framework is usually done on the planning activity. The other title for this activity is a definition of priorities. When the priorities are defined, they have to be implemented. Threats and risks to cybersecurity change constantly. Therefore, the reviewing of threats and risks must be done permanently. The decision can be made to improve cybersecurity after reviewing the threats and risks.

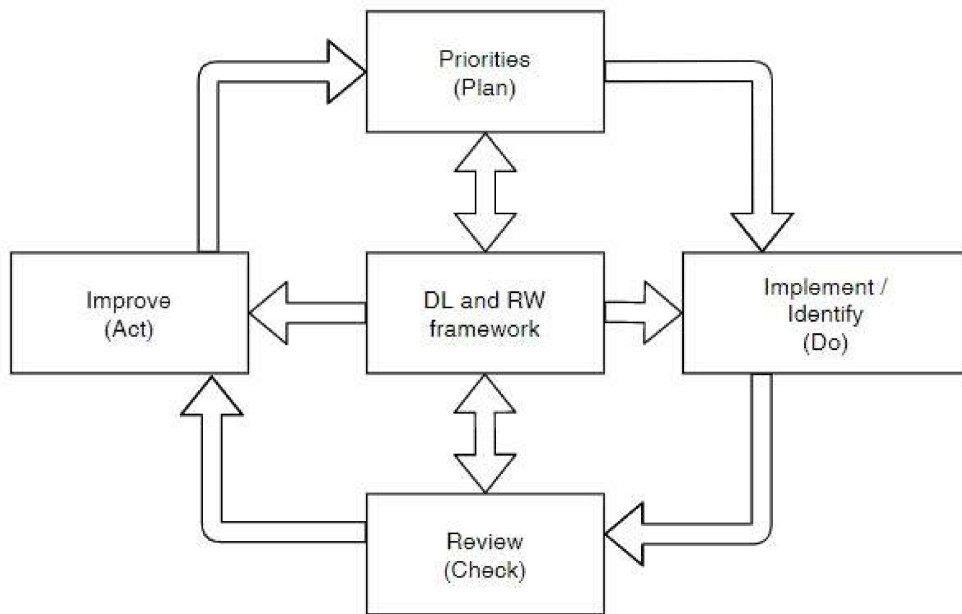

**Figure 5.** A framework of secure and reliable DL infrastructure.

It is also appropriate to lay down specific measures to create security appropriate to the user. As for the selection of tools, Table 2 enumerates domains of the application of cybersecurity and the technical measures applied to these domains.

**Table 2.** Domains of applications of a cybersecurity model.

| Domains of Applications of Cybersecurity Model | Actions | Technical Measures |
|---|---|---|
| AC—access control | Define requirements for system access | TK—control of privileged user |
| | Control the access to the system from inside | UV—firewall management |
| | Control remote access to the system | |
| | Allow access to the data only to those users and processes that have a real need | |
| AM—asset management | Identify and document resources | TS—network segmentation |
| | Monitor and manage resources | |
| AU—audit, and responsibility | Define requirements for log records | UV—firewall management |
| | Protect log records | SM—SIEM |
| | Monitor and manage log records | DP—data loss prevention |
| AT—awareness and training | Inform users about threats | MV—security training |
| | Organize security training and education | |
| CM—configuration management | Define the main (minimal, baseline) configurations | TP—application protection |
| | Manage configuration and updates | GT—endpoint protection |
| IA—identification, and authentication | Allow access to authenticated resources | TK—control of privileged user |
| IR—incident response | Prepare an incident management plan | AN—anomaly detection |
| | Implement incident management | PS—intrusion prevention system |
| | Test incident management | UV—firewall management |
| MA—maintenance | Implement maintenance | SM—SIEM |
| MP—media protection | Define and label media | KP—Malware prevention |
| | Protect and control media | GT—endpoint protection |
| | Protect media channels | UV—firewall management |

**Table 2.** *Cont.*

| Domains of Applications of Cybersecurity Model | Actions | Technical Measures |
|---|---|---|
| PS—personal security | Protect sensitive information | DP—data loss prevention |
| | | TK—control of privileged user |
| | | GT—endpoint protection |
| PE—physical environment | Restrict physical access | - |
| RE—recovery | Manage backup copies | TK—control of privileged user |
| | Manage continuity of information security | |
| RM—risk management | Define, assess, and manage risk | SM—SIEM |
| CA—control assessment | Prepare and manage a plan for information systems security | - |
| | Define and manage tools of control | PS—intrusion prevention system |
| | | SM—SIEM |
| SA—situation awareness | Implement monitoring of threats | SM—SIEM |
| | | TP—application protection |
| | | AN—anomaly detection |
| SC—system and communication | Define security requirements for systems and communication | BK—wireless access control |
| | Manage communication with information systems | PS—intrusion prevention system |
| | | AN—anomaly detection |
| | | EN—email and web browser protection |
| | | ZS—web protection |
| | | VT—VPN access control |
| SI—system integrity | Know and manage the flaws of the information system | PS—intrusion prevention system |
| | Identify malicious contents | KP—Malware prevention |
| | Monitor network and systems | SM—SIEM |
| | Implement enhanced protection of email | EN—email and web browser protection |

We provide the information of Table 2 in concise form using a matrix to ensure secure and reliable DL infrastructure within the organization (Table 3).

**Table 3.** Implementation matrix of DL infrastructure framework.

| Domains of Application | Appropriate Technical Measures | | | | | | | | | | | | | | |
|---|---|---|---|---|---|---|---|---|---|---|---|---|---|---|---|
| | TK | KP | AN | TP | DP | EN | GT | UV | PS | TS | SM | VT | ZS | BK | MV |
| AC | x | | | | | | | x | | | | | | | |
| AM | | | | | | | | | | x | | | | | |
| AU | | | | | x | | | x | | | x | | | | |
| AT | | | | | | | | | | | | | | | x |
| CM | | | | x | | | x | | | | | | | | |
| IA | x | | | | | | | | | | | | | | |
| IR | | | x | | | | | x | x | | | | | | |
| MA | | | | | | | | | | | x | | | | |
| MP | | x | | | | | x | x | | | | | | | |
| PS | | | | | x | | x | x | | | | | | | |
| PE | | | | | | | | | | | | | | | |
| RE | x | | | | | | | | | | | | | | |
| RM | | | | | | | | | | | x | | | | |
| CA | | | | | | | | | x | | x | | | | |
| SA | | | x | x | | | | | | | x | | | | |
| SC | | | x | | | x | | | x | | | x | x | x | |
| SI | | x | | | | x | | | x | | x | | | | |

The use of the secure and reliable DL infrastructure framework (Figure 5) and the description of the technical measures taken for distance learning at different IT security levels within the organization (Figure 4) in the field of cybersecurity can significantly increase the resilience to cyberattacks:

- Domains of the cybersecurity model: 17;
- Actions: 35;
- Technical measures for DL and RW: 39;
- Levels of IT security in the organization: (0–5).

The appropriate choice of technical measures and their prioritization, as shown in Figure 6, can ensure the full security of the educational institution's infrastructure during DL and RM. We can observe in Figure 6 that the specific technical measures are joint to the appropriate activities of the framework.

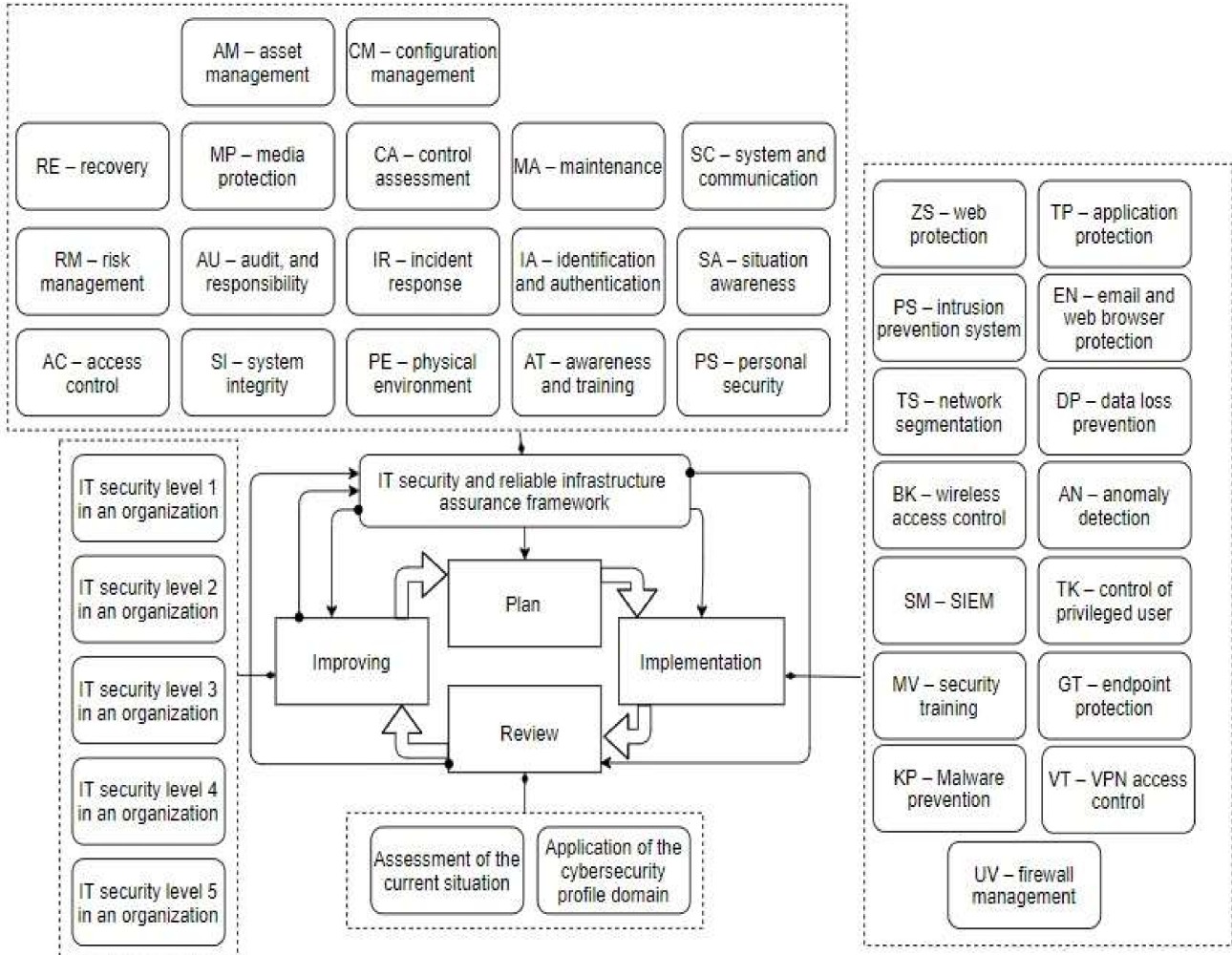

**Figure 6.** Secure DL infrastructure framework enriched by technical measures.

The technical measures to ensure cybersecurity may be selected for the domains that will be protected or that are of higher priority and subject to the proposed DL infrastructure framework matrix (Table 3). Capacity within the organization is also considered and is taken into account. As the level of maturity increases, the plan must be updated (Figure 5), followed by the implementation of measures and review of existing measures while ensuring the security level of DL infrastructure.

For the proper and successful application of the cybersecurity profile, we provide a matrix of ensuring secure and reliable DL infrastructure in the organization (Table 4). The matrix is used to assign the technical measures to the appropriate IT security levels.

**Table 4.** The level of assurance of secure and reliable IT infrastructure in the organization.

| Domains of Application | Levels of IT Security | | | | | |
|:---:|:---:|:---:|:---:|:---:|:---:|:---:|
| | 0 | 1 | 2 | 3 | 4 | 5 |
| AC | | x | x | x | x | x |
| AM | | | x | x | | |
| AU | | | x | x | x | x |
| AT | | | x | x | x | |
| CM | | | x | x | x | x |
| IA | | x | x | x | | |
| IR | Assessment of the current situation | | x | x | x | x |
| MA | | | x | x | | |
| MP | | x | x | x | | |
| PS | | | x | | | |
| PE | | x | x | x | | |
| RE | | | | x | x | x |
| RM | | | x | x | x | x |
| CA | | | x | x | x | |
| SA | | | | x | x | |
| SC | | x | x | x | x | x |
| SI | | x | x | x | x | x |

*4.4. A Case Study*

Each institution starts to take care of security from a level 0 to assess what measures it already has and can use. If such an institution decides to set level 1 of secure and reliable DL infrastructure, the institution uses the proposed framework (Figure 5) and reviews the domains of the application of the cybersecurity model. Level 1 of the cybersecurity model includes AC, IA, MP, PE, SC, and SI, as shown in Table 4. The technical measures for the implementation of security to the selected domains can be found in Table 3. They are as follows: TK, UV, KP, GT, AN, EN, PS, SM, VT, ZS, and BK. The joint information for the implementation of security level 1 is shown in Table 5.

**Table 5.** Implementation of level 1 to ensure secure and reliable IT infrastructure.

| Domains of Application of the Cybersecurity Model | DL and RW Technical Measures | | | | | | | | | | |
|---|---|---|---|---|---|---|---|---|---|---|---|
| AC | TK | | | | | UV | | | | | |
| IA | TK | | | | | | | | | | |
| MP | | KP | | | GT | UV | | | | | |
| PE | | | | | | | | | | | |
| SC | | | AN | EN | | | PS | | VT | ZS | BK |
| SI | | KP | | EN | | | PS | SM | | | |
| All measures of security level 1: | TK | KP | AN | EN | GT | UV | PS | SM | VT | ZS | BK |

### 4.5. Protection of Personal Data

Recently, the protection of personal data has received a lot of attention. Higher education institutions possess personal data, and therefore they have a statutory and regulatory responsibility for the careful processing of personal data. Personal data protection issues are particularly relevant during the implementation of DL and RW as personal data are not processed in a relatively secure intranet environment of the organization, but the data are transmitted and processed outside the intranet. Therefore, additional security measures are needed to ensure the secure transmission of personal data over the internet as well as the processing and storage of personal data whenever a person works remotely.

As our uses and approaches to technology generate new cybersecurity challenges, regulations are growing in number and complexity. This growth in regulatory changes often overwhelms organizations as they need to become more flexible to comply quickly and simultaneously with different mandatory controls and requirements. Personal data protection is specifically mapped with Control the privileged user rights (TK) to help organizations to meet multiple General Data Protection Regulation (GDPR) [17] and ISO 27001 [34] requirements with one comprehensive solution.

The implementation of the GDPR requirements is a complex process that requires considerable resources and expertise. Depending on the nature of the institution, its activities, the volume of data managed and processed, international and national organizations offer various methodologies and models that have been developed for the implementation of the GDPR. The State Data Protection Inspectorate of Lithuania has drawn up a 12-step set of recommendations to guide the preparation process for the implementation of the GDPR [38]. These recommendations are based on the guidelines of the UK Information Commissioner's Office [39], adapting them to the case of Lithuania. These steps are as follows: (1) awareness; (2) information that you are possessing; (3) provision of privacy information; (4) rights of data subjects; (5) implementation of the subject right to acquaint with personal data; 6) legal base of personal data management; (7) consent; (8) data of children; (9) violations of management of personal data; (10) applied personal data protection; (11) officer of personal data protection; and (12) internationalization.

We can recall that the institution must periodically review the security risks of IT infrastructure and improve protection when risks change. This is prescribed according to the secure and reliable IT framework (Figure 5). The same rule must be applied to the implementation of GDPR, as well.

### 4.6. Licensing of Software

When purchasing (whether for payment or free) software, you are entitled to use the program, but not to become the owner of the program. The terms and conditions regarding the use of the program depend on the license in which the copyright holder defines the conditions of use of the program in a particular case. The user of the program must understand and agree to and use the terms of the license without violating the existing restrictions. If the license terms are not accepted, the software cannot be used. Generally, a license entitles you to install a program on one computer and have a backup. The license may also include other authorizations, such as free use with additional restrictions, modification of the program, and distribution under certain conditions. There are two main types of licenses: free software and proprietary. Public domain software may be distinguished as a separate group. The four types of licenses are used the most frequently. They are as follows: (1) Copyright (©); (2) Creative Commons (CC); (3) Public Domain (PD); and (4) author-defined license.

Software used in the teaching/learning process requires a careful assessment of the software distribution and use license (terms and conditions) to avoid possible legal consequences or potential security breaches. Special care should be taken when using software that is free or temporarily free and that can be unlawfully distributed, with security issues, etc.



Digital content is usually shared using the Creative Commons (CC) license. Creative Commons is an international non-profit organization that has created several standard license agreements. The authors using these types of licenses can define terms of use of their products. Creative Commons defined four types of terms of product usage [40]. Combining these four types of terms six types of Creative Commons licenses and two types of Public Domain licenses are obtained. The terms of product usage are labeled using a special graphic character and/or a short description of terms of product usage. Creative Commons licenses enable usage of products without the separate permission of authors and/or holders of property rights. Different types of licenses define different terms of product usage. Creative Commons licenses apply practice that some property rights are reserved, meanwhile, the traditional copyright applies practice that all the property rights are reserved. Creative Commons licenses restrict only some rights of product usage.

The use of software that is copied, hacked, having an uncertain origin, or without specified authorship, in some cases, may be freely distributed; however, it may lead to certain risks, as well:

- Legal or financial consequences;
- The possible distribution of malicious software;
- Possible improper, illegal processing, collection (leakage) of personal data;
- Frequently has poor or no documentation at all;
- No guarantees are given regarding the consequences of this program usage;
- Usually, such a program is not supported, updated, bugs are not fixed, and technical assistance is not provided.

Although cybersecurity has always been important, it arguably has never been more vital for organizations to protect their data and reduce the risk of being hacked with so many of us working from home these days. The larger number of employees working from home leaves an organization open to more risks, as a home set-up will often be far less restricted compared to the office-based one. By consolidating IT, systems organizations can control what software their employees utilize, identify all assets (asset management (AM)), and protect against cyberattacks. The proposed model can be provided with the data for which assets are connecting to servers. Cybersecurity can protect the services (and remote access into the organization's infrastructure) by creating policies.

## 5. Results and Discussion

The proposed models are validated by experts in distance learning and IT security. The experts evaluated the models anonymously. The single qualification requirement for the experts was to have at least five years' experience of employment in the studies organization process, including distance learning and/or IT security. The experts were the distance learning process coordinators in their organizations, having a possibility to compare the use of IT infrastructure before and during the pandemic, as they first identified the challenges of the pandemic and had to assure a successful and secure study process.

Experts were invited to evaluate the proposed models of DL infrastructure and security. One of our co-authors is employed as a security manager of a whole university network. So, he practically knows the viability of the proposed models and their technical merits to ensure security of distance learning infrastructure. All the suggestions came from the practical point of view.

To ensure the anonymity of expert assessments, an expert survey was conducted using the anonymous survey tool Google Forms [41].

Because the proposed models are of an applied nature, a method based on Ikoma et al. [42] and the basic validation principles described in the IEEE 1012-2012 standard [43] are applied:

1. Compliance with the requirements of the product;
2. The usability of the product.

Model evaluation criteria and scale are based on the Likert methodology presented by McLeod [44]. The Likert scale was chosen to measure the expert opinions. According

to the Likert methodology, the questionnaire presents the question as a statement and several answer options. The options for an answer must show the extent to which the respondent agrees.

Five types of answers were used to assess the fulfillment of the requirements: "fulfilled", "more fulfilled than not", "neither fulfilled nor not", "more unfulfilled than fulfilled", and "unfulfilled". Their corresponding numeric values were 5, 4, 3, 2, and 1, respectively.

To assess the suitability of the model the following options were suggested: "suitable", "more suitable than not", "neither suitable nor unsuitable", "more unsuitable than suitable", and "unsuitable". Their corresponding numeric values were 5, 4, 3, 2, and 1, respectively.

The number of experts was chosen based on assumptions developed in the evaluation theory, which argues that the reliability of the aggregated solutions and the number of experts is linked to the factor determining the effectiveness of the research. Libby and Blashfield [45] have shown that the accuracy of decisions and assessments made by the group, consisting of 10 experts, is not inferior to that of a large expert group. The highest percentage of reliability is obtained with the evaluation of at least 7–10 experts, later the percentage of reliability changes very insignificantly, therefore 15 experts were invited to evaluate the validity of the distance learning models. The qualification of the experts was as follows: 14 of them had a doctoral degree, 1 had a master's degree, 10 of them were researchers in the field of technological sciences (9 of them were in the field of computer science, 1 in the field of mechanics), 5 of them were in the field of educational sciences; 6 of them were professors of universities (5 different universities), 3 of them were secondary school teachers, 5 of them were either higher or secondary school administrators, and 1 was a security expert from the business. Among the experts, eight female and seven male experts were present. Their age varied from 35 to 65 years. The mean age was 52.73 years. The standard deviation of the age was 8.99 years.

To assess the compliance of the models with the requirements, the relevant criteria for infrastructure and security models were formulated following the requirements. They are as follows:

1.  The infrastructure model includes the key components of DL and RW infrastructure.
2.  The security concept model covers: (1) IT infrastructure security profile, (2) levels of IT security, and (3) secure and reliable DL infrastructure framework.
3.  The matrix is the appropriate form for the planning of the implementation of the DL infrastructure framework, which joins the levels of IT security and IT infrastructure security profile.
4.  The security concept model includes the protection of personal data.
5.  The security concept model includes the management of copyright and licenses for digital content and software in education.

In total, 15 experts were invited to verify the validity of the models and 10 experts filled out anonymous evaluation questionnaires. Table 6 presents the data provided by the experts to assess the compliance of the models with the criteria.

According to the assessment data presented in Table 6, we can observe that five experts out of ten were not critical. They thought that the presented models fully satisfied all the raised requirements and assigned the highest assessment values possible. The least assessment values among all the presented values were assigned to criterion 2, which is used to assess the security concept model. The security concept model is presented in an abstract way. Five experts out of ten thought that such a presentation did not fully reveal what is intended by it. Two experts out of ten were quite critical. They thought that there is room for the improvement of all the models.

**Table 6.** Compliance of the models with the requirements.

| Experts | Criterion No 1 | Criterion No 2 | Criterion No 3 | Criterion No 4 | Criterion No 5 | Total |
|---|---|---|---|---|---|---|
| 1 | 4 | 4 | 4 | 4 | 4 | 20 |
| 2 | 4 | 4 | 4 | 4 | 4 | 20 |
| 3 | 4 | 4 | 4 | 5 | 5 | 22 |
| 4 | 4 | 4 | 5 | 5 | 5 | 23 |
| 5 | 5 | 4 | 5 | 5 | 5 | 24 |
| 6 | 5 | 5 | 5 | 5 | 5 | 25 |
| 7 | 5 | 5 | 5 | 5 | 5 | 25 |
| 8 | 5 | 5 | 5 | 5 | 5 | 25 |
| 9 | 5 | 5 | 5 | 5 | 5 | 25 |
| 10 | 5 | 5 | 5 | 5 | 5 | 25 |

The anonymous evaluation questionnaires for the usability of the models were completed by 10 experts out of 15 experts. The criteria for the evaluation were as follows:

1. The infrastructure model includes the key components of DL and RW infrastructure.
2. The security concept model covers: (1) IT infrastructure security profile, (2) levels of IT security, and (3) secure and reliable DL infrastructure framework.
3. The matrix is the appropriate form for the planning of the presentation of the implementation of the DL infrastructure framework, which joins the levels of IT security and IT infrastructure security profile.
4. The security concept model must include the protection of personal data.
5. The security concept model must include the management of copyright and licenses for digital content and software in education.

The assessment of the suitability of the models is presented in Table 7.

**Table 7.** Suitability of the models.

| Experts | Criterion No 1 | Criterion No 2 | Criterion No 3 | Criterion No 4 | Criterion No 5 | Total |
|---|---|---|---|---|---|---|
| 1 | 4 | 3 | 4 | 4 | 4 | 19 |
| 2 | 4 | 4 | 4 | 4 | 4 | 20 |
| 3 | 5 | 4 | 4 | 5 | 4 | 22 |
| 4 | 5 | 4 | 5 | 5 | 5 | 24 |
| 5 | 5 | 4 | 5 | 5 | 5 | 24 |
| 6 | 5 | 4 | 5 | 5 | 5 | 24 |
| 7 | 5 | 5 | 5 | 5 | 5 | 25 |
| 8 | 5 | 5 | 5 | 5 | 5 | 25 |
| 9 | 5 | 5 | 5 | 5 | 5 | 25 |
| 10 | 5 | 5 | 5 | 5 | 5 | 25 |

According to the assessment data presented in Table 7, we can observe that four experts out of ten were not critical. They were fully satisfied with the usability of the presented models and assigned the highest assessment values possible. The least assessment values among all the presented values were assigned to criterion 2 that is used to assess the suitability of the security concept model. The security concept model is presented in a quite abstract way. Six experts out of ten thought that such a presentation did not fully reveal what is intended by it. The first expert was especially critical in assigning the value 3 that means that the model was deemed "neither suitable nor unsuitable". Two experts out of ten were quite critical. They thought that there is room for the improvement of the suitability of all the models.

We can observe in Table 1 presented at the end of the review of related work that just two research works [15,16] are devoted to the consideration of the DL infrastructure and security together. However, both research works [15,16] devoted all attention to

the investigation of the DL infrastructure and security was mentioned at a very abstract level without providing any details how to ensure it. Other research works [20,25–27] summarized in Table 1 investigated only security. Different approaches were presented; however, no one approach investigated the technical measures required to implement security, and levels of IT security were not considered.

In this paper, we consider the security of DL infrastructure at the full length. We introduce a security concept model. We provide an IT infrastructure security profile, the levels of IT security, and a secure DL infrastructure framework. We enumerate the technical measures to implement the security concept model. We join the specific technical measures into appropriate activities of the framework. We show the technical measures to implement to achieve a higher security level if the institution is ready for it. We introduce an implementation of GDPR and software licensing into our security concept model. Just a few authors [28,29] discussed the problems of implementation of GDPR in the LMS and presented a possible solution. To the best of our knowledge, no author considered an implementation of GDPR as a component of the security model of the DL infrastructure. To the best of our knowledge, no author considered the problems of software licensing in any context. Consequently, we have presented the security model of the DL infrastructure, which covers all the possible domains related to the security of the DL infrastructure.

## 6. Conclusions

The proposed models are intended for higher education institutions whose lecturers had to adapt their teaching activities to the pandemic, integrate DL elements into their subject, and solve the new issues related to the security of the DL infrastructure. The proposed model of the DL infrastructure consists of the following components: user devices, network resources, virtualization, computing resources, data warehouses, data resources, software, continuity of activities, security, and network perimeter security.

The proposed model of the security of the DL infrastructure is presented as a hierarchical model consisting of two levels. The first level includes people, organizational processes, and security assurance. The security assurance is divided further (the second level) into the protection of personal data, copyright and licensing of digital content and software, and security of IT infrastructure. To ensure the security of IT infrastructure, an IT infrastructure security profile, the levels of IT security, and a secure DL infrastructure framework are provided. The technical measures to implement the desired security level of the institution were also presented.

The application of the proposed models will ensure security in the DL process for higher education institutions. During the assessment process, the experts decided that the proposed models fully fit the need of DL infrastructure, helping the administration to find the best solution on preparation and implementation of the DL processes. The proposed models of DL infrastructure and security fully meet the raised requirements and are suitable for use.

The first limitation of the study is that all the main contributing authors of the research to the methodology of the presentation have experience in administering university software systems. It is an advantage that the subject is well-known. On the other hand, it is a limitation since such knowledge presents an insider view. It lacks abstractions. To solve this limitation, future research should invite a co-author who would add to the study using abstractions for the representation of the subject. The second limitation of the study is that the validation by the experts was accomplished fully anonymously. It was not possible to collect information on the participating experts that would not disclose the person. For example, whether he or she is a representative of either teaching technologies or IT security, or whether he or she is a professor, administrator, or teacher. Then it would be possible to relate this information with their judgment and to decide where the possible weaknesses of the proposed models were. To solve this limitation, we suggest collecting some information, which would not disclose a person's identification to the participating experts.

The future direction of our research is the development of a model for general data protection regulation and licensing of digital content and software.

**Author Contributions:** Conceptualization, Š.G. and A.V.; Methodology, V.J. and D.G.; Validation, R.B. (Renata Burbaite) and A.V.; Formal analysis, R.B. (Rita Butkiene) and B.M.; Writing—Original Draft Preparation, Š.G. and D.G.; Writing—Review and Editing, V.J. and D.G.; Visualization, Š.G.; Project administration, A.V.; Funding Acquisition, A.V. and V.J. All authors have read and agreed to the published version of the manuscript.

**Funding:** This paper is supported in part by European Union's Horizon 2020 research and innovation program under Grant Agreement No. 830892, the project "Strategic programs for advanced research and technology in Europe" (SPARTA) and Lithuanian Research Council financed project "Model of distance working and learning organization and recommendations for extreme and transition period" (EKSTRE) (1 June 2020—31 December 2020). Grant Agreement S-COV-20-20.

**Institutional Review Board Statement:** Ethical review and approval were waived for this study, as this study involves no more than minimal risk to subjects.

**Informed Consent Statement:** Informed consent was obtained from all subjects involved in the study.

**Data Availability Statement:** The data presented in this study are available on request from the corresponding author. The data are not publicly available due to the data restriction policy by the grant provider.

**Conflicts of Interest:** The authors declare no conflict of interest.

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
