# Peer review of "Sustainable and Security Focused Multimodal Models for Distance Learning"

_sustainability, doi:10.3390/su14063414_

Round 1

Reviewer 1 Report

This paper proposes Sustainable and Security Focused Multimodal Models for Distance Learning. However, I have the following concerns:

  • The problem is not clear, authors introduce many issues that are not focused, and therefore the focus of this paper is clear.
  • The abstract is focused on the problem directly. It is written in very short statements that are not  connected together, therefore, I found it difficult to understand the problem exactly. I addition, the aim of the paper is also not concentrated. 
  • In Introduction also there no section that highlight the problem and the contribution of the paper is not mentioned.
  • It is not clear, what is the reference model that can be considered as a baseline for benchmarking the proposed models if they satisfy security issues or sustainability.
  • Authors are required to give the procedural definition of both security and sustainability at the beginning of the manuscript.
  • It is not clear how the authors evaluate the viability of the proposed models in terms of security and also sustainability. The experts evaluation of the frameworks are not always enough since those two terms should be handled also according to their technical merits.

I suggest authors refer to some studies in the literature that considered the technical aspects of security and sustainability with regards to distance learning.

Author Response

Dear 

please find attached requested updates for the paper

on behalf of the authors

Daina Gudoniene

Reviewer 2 Report

The paper is well structured and follows a rigid scientific methodology. The literature review is well executed and explained. The degree of novelty of the paper is not very high but it still adds a point of view on the academic debate
I suggest clarifying the research question in the first part of the paper.
The introduction of Distance Learning technologies is not new and was not caused by the pandemic. Some of the things that are presented as new are actually established practices.
I suggest updating the introductory part by considering the environments in which distance learning is a consolidated reality.

Author Response

(The authors gave the same response as above.)

Round 2

Reviewer 1 Report

The introduction section should be restructured.

The conclusion should be restated as well.

English editing is strongly required.

Author Response

Dear

please find updated paper. With truck changes you may see improvements in the Abstract and conclusions. 

thank you for all your comments 

regards, Daina Gudoniene